# O2O-Afford: Annotation-Free Large-Scale Object-Object Affordance Learning

**Kaichun Mo**[1], **Yuzhe Qin**[2], **Fanbo Xiang**[2], **Hao Su**[2], **Leonidas Guibas**[1]

[1]Stanford University   [2]UCSD

https://cs.stanford.edu/~kaichun/o2oafford

**Abstract:** Contrary to the vast literature in modeling, perceiving, and understanding agent-object (*e.g.* human-object, hand-object, robot-object) interaction in computer vision and robotics, very few past works have studied the task of object-object interaction, which also plays an important role in robotic manipulation and planning tasks. There is a rich space of object-object interaction scenarios in our daily life, such as placing an object on a messy tabletop, fitting an object inside a drawer, pushing an object using a tool, etc. In this paper, we propose a unified affordance learning framework to learn object-object interaction for various tasks. By constructing four object-object interaction task environments using physical simulation (SAPIEN) and thousands of ShapeNet models with rich geometric diversity, we are able to conduct large-scale object-object affordance learning without the need for human annotations or demonstrations. At the core of technical contribution, we propose an object-kernel point convolution network to reason about detailed interaction between two objects. Experiments on large-scale synthetic data and real-world data prove the effectiveness of the proposed approach.

**Keywords:** Object-object Affordance, Vision for Robotics, Large-scale Learning

## 1 Introduction

We humans accomplish everyday tasks by interacting with a wide range of objects. Besides mastering the skills of manipulating objects using our fingers (*e.g.* grasping [1]), we must also understand the rich space of object-object interactions. For instance, to put a book inside a bookshelf, not only do we need to pick up the book by our hand, to place it, we still have to figure out possible good slots — the distance between the two shelf boards must be larger than the book height, and the slot should afford the book at a suitable pose. For us humans, we can instantly form such understanding of object-object interactions after a glance of the scene. Theories in cognitive science [2] conjecture that this is because human beings have learned certain priors for object-object interactions based on the shape and functionality of objects. Can intelligent robot agents acquire similar priors and skills?

While there is a plethora of literature studying agent-object interaction, very few works have studied the important task of object-object interaction. In an earlier work, Sun *et al.* [3] proposed to use Bayesian network to model human-object-object interaction and performed experiments on a small-scale (six objects in total) labeled training data with relative motions of humans and the two objects. Another relevant work by Zhu *et al.* [4] studied the problem of tool manipulation, which is an important special case of object-object interaction, and proposed a learning framework given RGB-D scanned human and object demonstration sequences as supervision signals. Both works modeled object-object interaction in a small scale and trained the models with human annotations or demonstrations. In contrary, we propose a large-scale solution by learning from simulated interactions without any need for human annotations or demonstrations.

In this paper, we consider the problem of learning object-object interaction priors (abbreviated as the *O2O priors* for brevity). Particularly, in our setup, we consider an acting object that is directly manipulated by robot actuators, and a 3D scene that will be interacted upon. We are interested in encoding and predicting the set of feasible geometric relationships when the acting object is afforded by objects in the scene along accomplishing a certain specified task. One important usage of our

5th Conference on Robot Learning (CoRL 2021), London, UK.

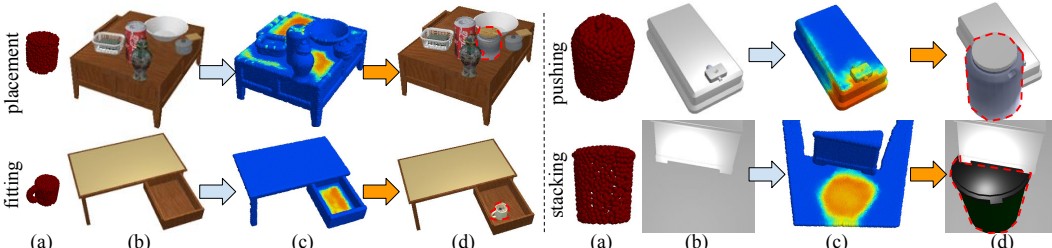

Figure 1: **The O2O-Afford Framework and Tasks.** For a given object-object interaction task (*i.e.* placement, fitting, pushing, stacking), our method takes as inputs a 3D acting object point cloud (a) and a partial 3D scan of the scene/object (b), and outputs an affordance prediction heatmap (c) that estimates the likelihood of the acting object successfully accomplishing the task at every pixel. At the test time, one may easily sample a position from the heatmap to perform the action (d).

*O2O priors* is to reduce the search space for downstream planning tasks. For example, to place a big object inside a cabinet having a few drawers with various sizes, the *O2O priors* may rule out those small drawers without any interaction trials and identify the ones big enough for the motion planners. Then, for the identified drawers, the *O2O priors* may further propose where the acting object could be placed by considering other factors, such as collision avoidance with existing objects in the drawers.

Next, we introduce how we encode the *O2O priors*. We formulate a per-point affordance labeling framework (Fig. 1) that unifies representations for various kinds of object-object interaction tasks. Given as input the acting object in different geometry, orientations and sizes, along with a partial 3D scan of an existing scene, we produce a point-wise affordance heatmap on the scene that measures the likelihood for each point on the scene point cloud of successfully accomplishing the task.

The affordance labels for supervision are generated by simulating the object-scene interaction process. Using the SAPIEN physical simulator [5] and large-scale 3D shape datasets [6, 7], we build up a large-scale learning-from-interaction benchmark that covers a rich space of object-object interaction scenarios. Fig. 1 illustrates the four diverse tasks we use in this work, in which different visual, geometric and dynamic attributes are essential to be learned for accurately modeling the task semantics. For example, to place a jar on a messy table, one need to find a flat tabletop area with enough space considering the volume of the jar; to fit a mug inside a drawer, the size and height of the drawer has to be big enough to contain the mug; etc.

As a core technical contribution, we propose an object-kernel point convolution network to reason about detailed geometric constraints between the acting object and the scene. We perform large-scale training and evaluation over 1,785 shapes from 18 object categories. Experiments prove that our proposed method learns effective features reasoning about the contact geometric details and task semantics, and show promising results not only on novel shapes from the training categories, but also over unseen object categories and real-world data.

In summary, our contributions are:

- we revisit the important problem of object-object interaction and propose a large-scale annotation-free learning-from-interaction solution;
- we propose a per-point affordance labeling framework, with an object-kernel point convolution network, to deal with various object-object interaction tasks;
- we build up four benchmarking environments with unified specifications using SAPIEN [5] and ShapeNet [6, 7] that covers various kinds of object-object interaction tasks;
- we show that the learned visual priors provide meaningful semantics and generalize well to novel shapes, unseen object categories, and real-world data.

## 2 Related Works

**Learning from Interaction.** Annotating training data has always been a heavy burden for supervised learning tasks. In robotics community, there has been growing interest in scaling up data collection via self-supervised interaction. This approach has been widely used to learn robot manipulation skills [8, 9, 10, 11], facilitate object representation learning [12, 13, 14], and improve the result of perception tasks, *e.g.* segmentation [15, 16], pose estimation [17]. However, collecting interaction

experience by a real robot is slow and even unsafe. A surrogate solution is physical simulation. Recent works explore the possibility of using data collected from simulation to train perception model [18, 19, 20]. By leveraging the interactive nature of physical simulation, researchers can also train network to reason object dynamics, *e.g.* mass [21], force [22], stability [23]. Benefits from the large-scale ShapeNet [6, 7] models, we can collect various type of interaction data from simulators.

**Agent-Object Interaction Affordance.** Agent-object interaction affordance describes how agents may interact with objects. The most common kind is grasping affordance. Recent works [24, 25, 26, 27, 28] formulate grasping as visual affordance detection problems that anticipate the success of a given grasp. An affordance detector predicts the graspable area from image or point cloud for robot grippers. Other works [29, 30, 31, 32] extend contact affordance from simple robot gripper to more complex human object interaction. However, most of the works require additional annotation as training data [24, 25, 29]. Recently, it was shown that visual affordances can also be reasoned from human demonstration videos in a weakly-supervised manner [33, 34]. Recent works [35, 36, 37] proposed automated methods for large-scale agent-object visual affordance learning.

**Object-Object Interaction Affordance.** Very few works have explored learning object-object interaction affordance. Sun *et al.* [3] built an object-object relationship model and associated it with a human action mode. It shows that the learned affordance is beneficial for downstream robotic manipulation tasks. Another line of works on object-object affordance focus on one particular object relationship: tool manipulation [4], object placement [38], pouring [39], and cooking [40]. However, all these works are performed on a limit number of objects. Most of these works also require human annotations or demonstrations. In our work, we conduct large-scale annotation-free affordance learning that covers various kinds of object-object interaction with diverse shapes and categories.

## 3 Problem Formulation

For every object-object interaction task, there are two 3D point cloud inputs: a scene partial scan $S \in \mathbb{R}^{n \times 3}$, and a complete acting object point cloud $O \in \mathbb{R}^{m \times 3}$, with center $c \in \mathbb{R}^3$, 1-DoF orientation $q \in [0, 2\pi)$ along the up-direction and an isotropic scale $\alpha \in \mathbb{R}$. The two point clouds are captured by the same camera and are both presented in the camera base coordinate frame, with the $z$-axis aligned with the up direction and the $x$-axis points to the forward direction. The output of our O2O-Afford tasks is a point-wise affordance heatmap $A \in [0, 1]^m$ for every point in the scene point cloud, indicating the likelihood of the acting object successfully interacting with the scene at every position. Fig. 1 shows example inputs and outputs for four different tasks.

## 4 Task Definition and Data Generation

As summarized in Fig. 1, we consider four object-object interaction tasks: placement, fitting, pushing, stacking. While *placement* and *stacking* are commonly used in manipulation benchmark [41, 42, 43, 44], the *fitting* and *pushing* tasks are also interesting for bin packing [45] and tool manipulation applications [46]. One may create more task environments depending on downstream applications. Although having different task semantics and requiring learning of distinctive geometric, semantic, or dynamic attributes, we are able to unify the task specifications to share the same framework.

### 4.1 Unified Task Environment Framework

**Task Initialization and Inputs.** Each task starts with creating a static scene, including one or many randomly selected ShapeNet models and a possible ground floor. Some objects in the scene may have articulated parts with certain starting part poses, depending on different tasks. The scene objects may be fixed to be always static (*e.g.* when we assume a cabinet is very heavy), or dynamic but of zero velocity at the beginning of the simulation (*e.g.* for an object on the ground to be pushed). In all cases, a camera takes a single snapshot of the scene to obtain a scene partial 3D scan $S$ as the input to the problem. Fig. 2 (a) illustrate example initialization scenarios in the four task environments. To interact with the scene, we then randomly fetch an acting object and initialize it with random orientation and size. We provide a complete 3D point cloud $O$, in the same camera coordinate frame as the scene point cloud, as another input to the problem.

*O2O Priors* **Parametrization.** For each task $\mathfrak{T}$, we define the *O2O priors* of an acting object $O$ interacting with a scene geometry $S$ as a per-point affordance heatmap $A_{\mathfrak{T}} \in [0, 1]^m$ over the scene point cloud $S \in \mathbb{R}^{m \times 3}$. For each point $p_i \in S$, we predict a likelihood $a_i \in [0, 1]$ of the acting object $O$

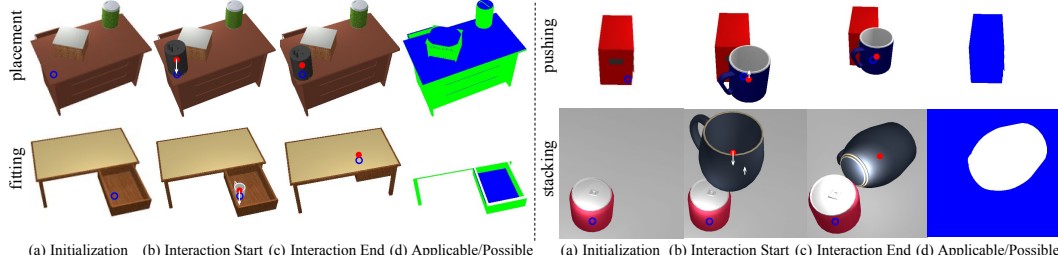

Figure 2: **Task Environment Specifications and Trajectory Illustrations.** For each task environment, from left to right, we respectively show the static scene for initialization (a), the state of objects at the beginning of interaction (b), the state of objects at the end of interaction (c), as well as the applicable (green+blue) and possible (blue) regions (d). We mark the interacting positions $p$ in blue circles and track the moving object centers $c$ using red dots. The white arrows used in (b) indicate the object trajectory moving directions: forward direction for *pushing*; gravity direction for others.

successfully interacting with the scene $S$ at position $p_i$ following a parametrized short-term trajectory $\tau_{\mathfrak{T}}(p_i)$, which is a hard-coded task-specific trajectory with the acting object center initialized at $c_i = p_i + r_{\mathfrak{T}}(O)$. Here, $r_{\mathfrak{T}}(O) \in \mathbb{R}^3$ is a task-specific offset that also depends on the acting object $O$. Fig. 2 (b) illustrates the starting states of the acting and scene objects, where red dots track the object centers $c_i$'s and blue circles mark the interacting position $p_i$'s, from which one may imagine the offsets $r_i = c_i - p_i$ for different tasks. We will define the task-specific offsets in Sec. 4.2.

**Simulated Interaction Trajectory.** For each interaction trial, we execute a hard-coded short-term trajectory $\tau_{\mathfrak{T}}(p_i)$ to simulate the interaction between the acting object $O$ and the scene objects $S$ at positition $p_i$. Every motion trajectory is very short, *e.g.* taking place within $< 0.1$ unit length, so that it can be preceded by long-term task trajectories, to make the learned visual priors possibly useful for many downstream tasks. The white arrows in Fig. 2 (b) show the trajectory moving directions – the forward direction for *pushing* and the gravity direction for the other three environments. The executed hard-coded trajectories are always along straight lines, though the final object state trajectories may be of free forms due to the object collisions. Fig. 2 (c) present some example ending object states.

**Applicable and Possible Regions.** For every simulated interaction trial, we randomly pick $p$ over the regions where the task is applicable and possible to succeed. Some scene points may not be applicable for a specific task. For example, for the *stacking* task, only points on the ground are applicable since we have to put the acting object on the floor for the scene object to stack over. Among applicable points, we only try the positions that are possible to be successfully interacted and directly mark the impossible points as failed interactions. For example, impossible points include the positions whose normal directions are not nearly facing upwards for the *placement* and *fitting* tasks. Fig. 2 (d) illustrate example applicable and possible masks over the input scene geometry.

**Metrics and Outcomes.** For each interaction trial, the outcome could be either successful or failed, measured by task-specific metrics. The metric measures if the intended task semantics has been accomplished, by detecting state changes of the acting and scene objects during the interaction and at the end. For example, in Fig. 2 (c), the *fitting* example shows that the drawer will be driven to close to check if the acting object can be fitted inside the drawer. See Sec. 4.2 for detailed definitions.

## 4.2 Four Task Environments

**Placement.** Each scene is initialized with a static root object (*e.g.* a table) with 0∼15 movable small item objects randomly placed on the root object to simulate a messy tabletop. The root object may have articulated parts, which are initialized to be closed or of a random starting pose with equal probabilities. The acting object is another small item object to be placed. All points are applicable on the scene, but only the positions with normal directions that are close enough to the world up-direction are possible. For the acting object center at start, we have an up-directional offset $r_z = s_z/2 + 0.01$, where $s_z$ is the up-directional object size, so that the acting object is 0.01 unit length away from contacting the intended interaction position $p$. The motion trajectory is along the gravity direction. The metric for success is: 1) the acting object has no collision at start; 2) the acting object finally stays on the countertop; and 3) the acting object drops off stably with no big orientation change.

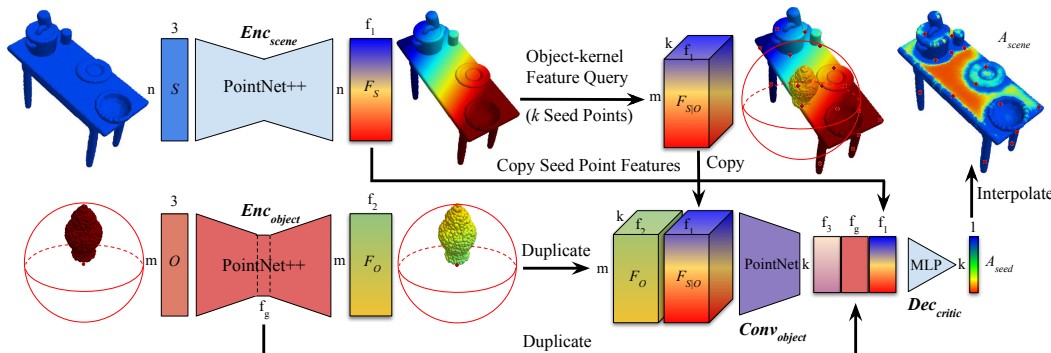

Figure 3: **Network Architecture.** Taking as inputs a partial 3D scan of the scene $S$ (dark blue) and a complete 3D point cloud of acting object $O$ (dark red), our network learns to extract per-point features on both inputs, correlate the two point cloud feature maps using an object-kernel point convolution, and finally predict a point-wise affordance heatmap over the scene point cloud.

**Fitting.** The scene contains only one static root object with articulated parts (*e.g.*, doors, drawers). At least one articulated part is randomly opened, while the other parts may be closed or randomly opened with equal probabilities. The acting object is a small item object to be fitted inside the drawer or shelf board. All points except the countertop points are applicable, since placing the item on countertop is concerned by the *placement* task. The positions with normal directions close enough to the world up-direction are possible. The starting acting object center and the motion trajectory are the same as in the *placement* task. Besides the three criteria for *placement*, there is one additional checking point for the metric: the door or drawer can be closed containing the acting object without being blocked.

**Pushing.** The scene object is dynamically placed on an invisible ground, initialized with zero velocity, and guaranteed to be stable by itself. There is no part articulation allowed in this task. The acting object is a small item object to push the scene object. All points on the scene are applicable and possible. The starting acting object center has offsets $r_x = -s_x/2 - 0.1, r_z = s_z/2 + 0.02$ where $s_x$ and $s_z$ are respectively the forward-directional and up-directional object sizes. The acting object moves along the forward-direction to push the scene object. The metric for success is: 1) the acting object has no collision at start; 2) there is a big enough motion of the scene object; 3) the actual moving direction is within $30°$ aligned with the forward-direction; and 4) the scene object does not topple.

**Stacking.** We first place a dynamic acting object stably with zero velocity on a visible ground. Then, we pick a second item as the scene object to stack over the acting object. We simulate the stacking interaction that drops the scene object on top of the acting object. For the cases that the two objects finally touch each other and stay stably on the ground, we consider a stacking task based on the final two object states. The scene object is initialized at the final stably stacking pose. All points on the ground are applicable and possible. The acting object center has an up-directional offset $r_z = s_z/2$, where $s_z$ is the up-directional object size. The metric for success is that the acting object has no big pose (center+orientation) change before and after the stacking interaction.

## 5 Method for Affordance Prediction

We propose a unified 3D point-based method, with an object-kernel point convolution network, to tackle the various O2O-Afford tasks. Though very simple, this method is quite effective and efficient in reasoning about the detailed geometric contacts and constraints.

### 5.1 Network Architecture

Fig. 3 illustrates the proposed pipeline. We describe each network module in details below.

**Feature Extraction Backbones.** We employ two segmentation-version PointNet++ [47] to extract per-point feature maps for the two input point clouds. We first normalize them to be zero-centered. For the 3D partial scene $S \in \mathbb{R}^{n \times 3}$, we train a **Enc**$_{\text{scene}}$ PointNet++ to extract per-point feature map $F_S \in \mathbb{R}^{n \times f_1}$. For the 3D acting object $O \in \mathbb{R}^{m \times 3}$, we use another **Enc**$_{\text{object}}$ PointNet++ to obtain per-point feature map $F_O \in \mathbb{R}^{m \times f_2}$ and a global feature for the acting object $F_g \in \mathbb{R}^{f_g}$.

**Object-kernel Point Convolution.** Our O2O-Afford tasks require reasoning about the contact geometric constraints between the two input point clouds. Thus, we design an object-kernel point

convolution module that uses the acting object as an explicit object kernel to slide over a subsampled scene seed points and performs point convolution operation to aggregate per-point features between the acting object and the scene inputs. This design shares a similar spirit to the recently proposed Transporter networks [48], but we carefully curate it for the 3D point cloud convolution setting. One may think of a naive alternative of simply concatenating two point clouds together at every seed point and training a classifier. However, this is computationally too expensive due to several forwarding passes over the two input points clouds with the acting object positioned at different seed locations.

Concretely, on the scene partial point cloud input $S \in \mathbb{R}^{n \times 3}$, we first sample $k$ seed points $\{p_1, p_2, \cdots, p_k\} \subset S$ using Furthest Point Sampling (FPS). Then, we move the acting object point cloud $O \in \mathbb{R}^{m \times 3}$, as an explicit point query kernel, over each of the sampled seed point $p_i$ to query a scene feature map $F_{S|O,p_i} \in \mathbb{R}^{m \times f_1}$ over the acting object points $O$. In more details, for each point $o_j \in O$, we query the scene feature at the position $o_j^i = o_j + p_i$ using the inverse distance weighted interpolation [47] in the scene point cloud $S \in \mathbb{R}^{n \times 3}$ with feature map $F_S \in \mathbb{R}^{n \times f_1}$. We query $t$ nearest neighbor points $\{e_1, e_2, \cdots, e_t\} \subset S$ to any $o$ ($i, j$ omitted for simplicity), and compute $F_{S|o}$ with

$$F_{S|o} = \frac{\sum_{l=1}^{t} w_l F_{S|e_l}}{\sum_{l=1}^{t} w_l}, w_l = \frac{1}{\|o - e_l\|_2}, l = 1, 2, \cdots, t, \tag{1}$$

where $F_{S|e_l} \in \mathbb{R}^{f_1}$ is the computed scene feature at point $e_l$. We aggregate all interpolated scene features $\{F_{S|o_j^i} \in \mathbb{R}^{f_1}\}_j$ over the acting object points $\{o_j^i\}_{j=1,2,\cdots,m}$ to obtain a final feature map $F_{S|O,p_i} \in \mathbb{R}^{m \times f_1}$ at every scene seed point $p_i$.

Concatenating the acting object feature map $F_O \in \mathbb{R}^{m \times f_2}$ and the interpolated scene feature map $F_{S|O,p_i} \in \mathbb{R}^{m \times f_1}$, we obtain an aggregated feature map $F_{SO|O,p_i} \in \mathbb{R}^{m \times (f_1 + f_2)}$ at every scene seed point $p_i$. We then implement the object-kernel point convolution $\mathbf{Conv_{object}}$ using PointNet [49] to obtain seed point features $F_{SO|p_i} \in \mathbb{R}^{f_3}$. The PointNet is composed of a per-point Multilayer Perceptron (MLP) transforming every individual point feature and a final max-pooling over all $m$ points.

**Point-wise Affordance Predictions.** For each scene seed point $p_i$, we aggregate the information of the computed object-kernel point convolution feature $F_{SO|p_i} \in \mathbb{R}^{f_3}$, the local scene point feature $F_{S|p_i} \in \mathbb{R}^{f_1}$, and the global acting object feature $F_g \in \mathbb{R}^{f_g}$, and feed them through $\mathbf{Dec_{critic}}$, implemented as an MLP followed by a Sigmoid activation function, to obtain an affordance labeling $a_{p_i} \in [0, 1]$, with bigger value indicates higher likelihood of a successful interaction between the acting object and the scene at $p_i$. After computing the per-point affordance labeling for the $k$ subsampled seed points, we interpolate back to all the locations in the scene point cloud, using the inverse distance weighted average, to obtain a final per-point affordance labeling $A_{scene} \in [0, 1]^n$.

## 5.2 Training and Loss

The whole pipeline is trained in an end-to-end fashion, supervised by the simulated interaction trials with successful or failed outcomes. The scene and acting objects are selected randomly from the training data of the training object categories. We equally sample data from different object categories to address the data imbalance issue. We empirically find that having enough positive data samples (at least 20,000 for task) are essential for a successful training. We train individual networks for different tasks and use the standard binary cross entropy loss. We use $n = 10000$, $m = 1000$, $k = 1000$, $t = 3$, and $f_1 = f_2 = f_3 = f_g = 128$ in the experiments. See supplementary for more training details.

## 6 Experiments

We use the SAPIEN physical simulator [5], equipped with ShapeNet [6] and PartNet [7] models, to do the experiments. We evaluate our proposed pipeline and provide quantitative comparisons to three baselines. Experiments show that we successfully learned visual priors of object-object interaction affordance for various O2O-Afford tasks, and the learned representations generalize well to novel shapes, unseen object categories, and real-world data.

### 6.1 Data and Settings

Our experiments use 1,785 ShapeNet [6] models in total, covering 18 commonly seen indoor object categories. We randomly split the different object categories into 12 training ones and 6 test ones. Furthermore, the shapes in the training categories are separated into training and test shapes. In total,

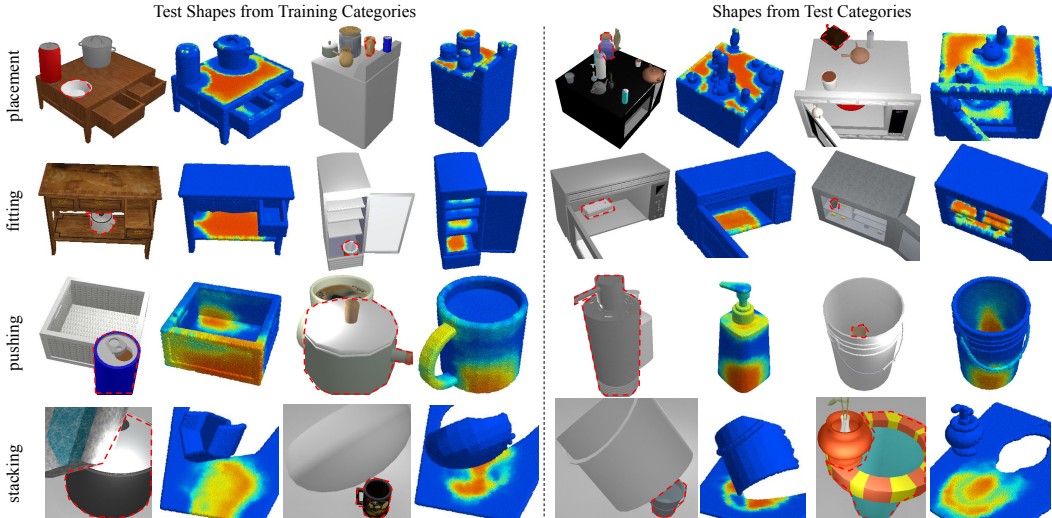

Figure 4: **Qualitative Results.** For each of the tasks, we show examples of our network predictions. Left two examples show test shapes from the training categories, while two right ones are shapes from the test categories. In each pair of result figures, we draw the scene geometry together with the acting object (marked with red dashed boundary) on the left, and show our predicted per-point affordance heatmaps on the right.

| | | F-score (%) | AP (%) | | | F-score (%) | AP (%) |
|---|---|---|---|---|---|---|---|
| placement | B-PosNor | 62.1 / 81.7 | 60.5 / 78.2 | pushing | B-PosNor | 31.9 / 34.9 | 37.0 / 35.5 |
| | B-Bbox | 80.9 / **90.6** | 90.5 / 94.5 | | B-Bbox | 33.2 / 35.0 | 39.2 / 37.6 |
| | B-3Branch | 63.8 / 77.1 | 69.8 / 82.3 | | B-3Branch | 35.2 / 36.6 | 42.2 / 36.4 |
| | Ours | **81.4** / 90.0 | **91.1 / 95.2** | | Ours | **35.5 / 40.3** | **46.9 / 43.1** |
| fitting | B-PosNor | 45.4 / 59.3 | 46.8 / 66.7 | stacking | B-PosNor | 79.3 / 77.9 | 79.9 / 76.5 |
| | B-Bbox | 69.5 / 79.5 | 80.1 / 80.6 | | B-Bbox | 85.7 / 83.2 | 87.7 / 87.2 |
| | B-3Branch | 48.2 / 56.9 | 47.1 / 60.7 | | B-3Branch | 87.3 / 84.8 | 90.8 / 88.2 |
| | Ours | **73.6 / 80.3** | **80.1 / 86.3** | | Ours | **89.6 / 87.5** | **91.7 / 90.8** |

Table 1: **Quantitative Evaluations.** We compare to three baselines **B-PosNor**, **B-Bbox**, and **B-3Branch**. In each entry, we report evaluations over test shapes from the training categories (before slash) and shapes in the test categories (after slash). Higher numbers indicate better performance.

there are 867 training shapes from the training categories, 281 test shapes from the training categories, and 637 shapes from the test categories. During training, all the networks are trained on the same split of training shapes from the training categories. We then evaluate and compare the methods by evaluating on the test shapes from the training categories, to test the performance on novel shapes from known categories, and the shapes in the test categories, to measure how well the learned visual representations generalize to totally unseen object categories. See supplementary for more details.

### 6.2 Baselines and Metric

We compare to three baseline methods **B-PosNor**, **B-Bbox**, and **B-3Branch**. **B-PosNor** replaces the per-point scene feature with 3-dim position and 3-dim ground-truth normal, while **B-Bbox** uses a 6-dim axis-aligned bounding box extents to replace the acting object geometry input. We compare to these two baselines to validate that the extracted scene features contain more information than simple normal directions and that object geometry matters. **B-3Branch** implements a naive baseline that employs two PointNet++ branches to process the acting object and scene point clouds as well as an additional branch taking as input the seed point position. Comparison to this baseline can help illustrate the necessity of correlating the two input point clouds. We use a success threshold 0.5 and employ two commonly used metrics: *F-score* and *Average-Precision (AP)*.

### 6.3 Results and Analysis

Table 1 shows the quantitative evaluations and comparisons. It is clear to see that our method performs better than the three baselines in most entries. We visualize our network predictions in Fig. 4, where we observe meaningful per-point affordance labeling heatmaps on both test shapes from the training categories and shapes in unseen test categories. For *placement*, the network learns to not only find

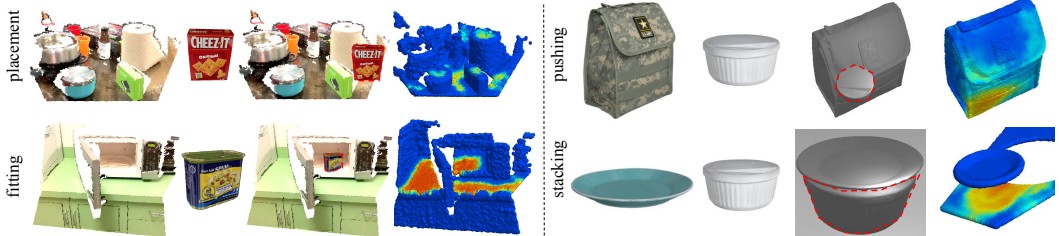

Figure 5: **Results on Real-world Data.** From left to right: we respectively show the scene geometry (a partial 3D scan from the camera viewpoint), the acting object (a complete 3D point cloud), the interaction snapshot (to illustrate the poses and sizes of the objects), and our point-wise affordance predictions. We scan noisy 3D partial scenes using an iPad Pro and choose acting objects from the YCB Dataset [50] in the *placement* and *fitting* examples, while evaluating over three Google Scanned Objects [51, 52, 53] in the rest. In the *fitting* case, only the predictions within the microwave are desired since the network is trained and expected to be tested over clean single scene objects.

|  |  | F-score (%) | AP (%) |
|---|---|---|---|
| placement | Ablated | **82.2 / 91.3** | 90.0 / **95.3** |
|  | Ours | 81.4 / 90.0 | **91.1** / 95.2 |
| fitting | Ablated | 68.0 / 78.3 | 77.8 / 84.2 |
|  | Ours | **73.6 / 80.3** | **80.1 / 86.3** |
| pushing | Ablated | 34.9 / 38.6 | 40.5 / 39.5 |
|  | Ours | **35.5 / 40.3** | **46.9 / 43.1** |
| stacking | Ablated | 82.6 / 80.4 | 87.2 / 83.5 |
|  | Ours | **89.6 / 87.5** | **91.7 / 90.8** |

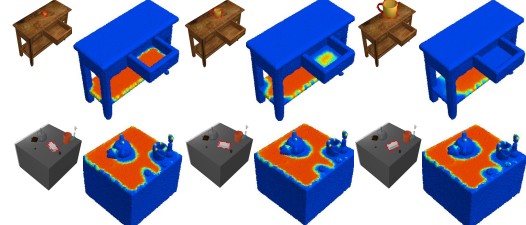

Table 2: **Ablation Study.** We compare to an ablated version that removes the computed object-kernel point convolution features $F_{SO|p_i}$.

Figure 6: **Result Analysis.** Our network predictions are sensitive to the acting object size (top-row) and orientation (bottom-row) changes.

the flat surface, but also avoid collisions from the existing objects. We observe similar patterns for *fitting*, with one additional learned constraint to find two shelves with enough height. For *pushing*, one should push an object in the middle to cause big enough motions and at the bottom to avoid toppling. For *stacking*, our network successfully learns where to place the acting object on the ground for the scene object to stack over. See supplementary for more results.

We also perform an ablation study in Table 2 to prove the effectiveness of the proposed object-kernel point convolution. We further illustrate in Fig. 6 that our network predictions are sensitive to the acting object size and orientation changes. In the top-row *fitting* example, increasing the size of the mug reduces the chance of putting it inside the drawer, as the drawer cannot be further closed containing a big mug. For the bottom-row *placement* example, we observe some detailed affordance heatmap changes while we rotate the cuboid-shaped acting object.

We directly try to apply our network trained on synthetic data to real-world 3D scans. Fig. 5 and Fig. G.3 in the supplementary shows some qualitative results. We observe that, though trained on synthetic data only, our network transfers to real-world collected data to reasonable degrees.

## 7   Conclusion

We revisited the important but underexplored problem of visual affordance learning for object-object interaction. Using state-of-the-art physical simulation and the available large-scale 3D shape datasets, we proposed a learning-from-interaction framework that automates object-object interaction affordance learning without the need of having any human annotations or demonstrations. Experiments show that we successfully learned visual affordance priors that generalize well to novel shapes, unseen object categories, and real-world data.

**Limitations and Future Works.** First, our method assumes uniform density for all the objects. Future works may annotate such physical attributes for more accurate results. Second, we train separate networks for different tasks. Future study could think of a way for joint training, as many features may be shared across tasks. Third, there are many more kinds of object-object interaction that we have not included in this paper. People may extend the framework to cope with more tasks.

## Acknowledgements

This research was supported by NSF grant IIS-1763268, NSF grant RI-1763268, a grant from the Toyota Research Institute University 2.0 program[1], a Vannevar Bush faculty fellowship, and gift money from Qualcomm. This work was also supported by AWS Machine Learning Awards Program.

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
