# OpenReview forum: "O2O-Afford: Annotation-Free Large-Scale Object-Object Affordance Learning"
_robot-learning.org/CoRL/2021/Conference — CoRL2021 Poster_

### Official Review · Reviewer_4Nfi · 2021-07-05

**Originality:** Good
**Technical Quality:** Excellent
**Clarity Of Presentation:** Very Good
**Impact:** 3

**Recommendation:**

Weak Accept: I recommend accepting the paper, but will not argue for my recommendation if the majority of other reviewers have a different opinion.

**Summary:**

This submission develops a method to populate a receptacle object pointcloud with a heatmap of stable resting points for a patient object being manipulated. A physics simulation determines how the two objects interact against a single physical force (either gravity applied to both objects or a "pushing" force coming off of one object). Four such binary "affordances" between objects are investigated: placement, pushing, fitting, and stacking.

*Post-response*

The authors have addressed my comments and questions, and clarified what I had previously thought was the main weakness of the paper was actually my own misunderstanding / lack of digging into the supp. The revised paper makes it clear that the method was applied to real world scans as well.

**Issues:**

Questions (ordered by priority):
Q1 Does the heatmap prediction generalize to real images of target receptacles and patient objects?

Q2 Table 1: usually only small gains over B-Bbox. Intuition for why this performs so well and whether proposed algorithm can be demonstrated much better on some other objects or tasks?

Q3 Are point clouds only what's visible to the camera as described in S3, or full 3D model pointclouds? I understood that just what the camera can see is "visible", but since PointNet can eat full 3D models, I wanted to clarify.

Q4 Was there any gravity force during "pushing" simulations?

Nit:
- 4.1 "with an arbitrary orientation and size"; didn't follow use of the word "arbitrary" here. Does this mean object default orientation and size were used without any randomization? Just trying to say model is agnostic to these values?

**Reviewer Expertise:**

Good: General knowledge of the area

**Strengths And Weaknesses:**

The main strength of the paper is the ability to label data without human annotation by leveraging a physics simulation and many object models. Rather than having humans describe, for example, whether a patient object comes to rest on top of or inside of a receptacle [as in Scalise et. al. "Improving Robot Success Detection using Static Object Data" IROS 2019] for the "stacking" and "fitting" actions, the submission uses a combination of environmental forces and condition states to define whether something has been successfully placed, pushed, fitted, or stacked. For example, for "fitting", the patient must rest inside the receptacle, which is then 'closed' [drawers, etc. on articulated object models] to ensure no collisions.

The main weakness of the paper is that it does not investigate whether the heatmap prediction, which is learned only in simulation over a small set of ShapeNet categories, transfers to much noisier, real world pointclouds. Ideally, such an evaluation would also demonstrate whether inferred placing/pushing/fitting/stacking locations could be acted out via a robot gripper. Minimally, though, we'd like to see qualitatively whether the heatmap prediction can be applied in the real world, given the large sim to real gap between background-less ShapeNet objects and real world clutter.

**Summary Of Recommendation:**

The work is technically sound and well-motivated as contribution to the vision community. Without demonstrating that the heatmap prediction method can transfer to real world camera input (minimally) and be used to inform a robots placing and pushing ability (maximally), it is difficult to assess the impact of the work in simulation on uniform density 3D models alone.

---

### Official Review · Reviewer_dU8R · 2021-07-23

**Originality:** Good
**Technical Quality:** Good
**Clarity Of Presentation:** Very Good
**Impact:** 3

**Recommendation:**

Strong Accept: I recommend accepting the paper and will argue for my recommendation even if other reviewers hold a different opinion.

**Summary:**

This paper proposes a data-driven framework to learn object-to-object affordances, ie. the areas where an acting object and a scene object can interact with one another. First, a dataset of affordances is collected via a self-supervised process, where objects in a simulated environment interact with each other at random locations and the resulting affordance success label is recorded. Given this dataset of affordances, a network is trained to process 3d point clouds of the objects using an object-kernel point convolution operation and outputs the affordance heatmap over the scene object. Experiments demonstrate that the proposed approach can learn affordances for four different types of object interaction with relatively high accuracy and can generalize to unseen object categories and real-word data.

**Issues:**

* more complete related works section
* revise “object-kernel point convolution” section to be easier to follow
* discuss runtime analysis, memory usage, batch size, training time, etc
* discuss alternative architectures, eg. transformer-style methods. If possible, add them as baselines
* (if possible) present quantitative results on the real-world results dataset
* expand on how the ablated version of the method is different from the original method

**Reviewer Expertise:**

Good: General knowledge of the area

**Strengths And Weaknesses:**

This paper addresses an important problem (ie. learning where objects can interact) and motivates the problem well in the introduction. The related works section also summarizes much of the relevant work well. There are however a few additional references worth discussing to provide full context of related works:
* Where2Act: From Pixels to Actions for Articulated 3D Objects, by Mo et al.
* Dexterous Robotic Grasping with Object-Centric Visual Affordances, by Mandikal and Grauman
* Learning Affordance Landscapes for Interaction Exploration in 3D Environments, by Nagarajan and Grauman

Generally, the methods section and experiments section are well-written. In particular, section 3 (Problem Formulation) sets up the overall problem clearly and the experiments section is well-structured. There are however several areas for improvement regarding clarity and completeness. First, I found the “object-kernel point convolution” section to be a bit dense; simplifying the notation and adding additional explanations to contextualize the technical details can help to clarify this section. Furthermore, it would be helpful to add a discussion on runtime and memory analysis to understand the efficiency of the proposed learning algorithm. This is especially important for processing 3d data, which typically entails resource-intensive solutions. Regarding the experiments section, the results are compelling but the baselines are sparse. Given that several papers have proposed special-purpose architecture to learn on 3d data / affordances, and also considering the recent emergence of transformer-based architectures, it would help for the paper to discuss these architectures and to possibly adopt some of them as baselines. Apart from this the results and analysis section is strong; in particular figure 6 validates that the framework is receptive to the geometry of objects. One suggestion for improvement is to present quantitative results on the real-world data (if possible) and to clarify the distinction between the ablated and original method.

**Summary Of Recommendation:**

Overall this is a good paper, because it (1) focuses on an important problem, (2) presents a scalable approach to address the problem, and (3) is well-written. In particular, the fact that the affordance data can be collected at scale in simulation and that the resulting model appears to generalize to real-world data makes this a compelling paper. There are still several areas for improvement, primarily concerning the baselines, runtime analysis, and clarity in describing some components. See the comments below.

**Update**: changing recommendation to strong accept. The author rebuttal addresses many of my concerns; also the rebuttal response to other reviewers seems to satisfy concerns raised by other reviewers as well.

**@AC: The AC should however note that this is not my immediate area of expertise, so the recommendation of other reviewers who have more expertise should be weighed more than mine.**

---

### Official Review · Reviewer_esX6 · 2021-07-24

**Originality:** Very Good
**Technical Quality:** Good
**Clarity Of Presentation:** Excellent
**Impact:** 4

**Recommendation:**

Strong Accept: I recommend accepting the paper and will argue for my recommendation even if other reviewers hold a different opinion.

**Summary:**

*This work performs a large-scale study on the object-object affordance problem. The proposed learning-from-interaction in simulation avoids expensive real world data collection or human annotation.

*It proposes a per-point affordance labeling framework, with an object-kernel point convolution network, to deal with various object-object interaction tasks.

*It builds up four benchmarking environments with unified specifications using SAPIEN and ShapeNet  that covers various kinds of object-object interaction tasks.


**Issues:**

The baseline methods of B-Normal and B-Bbox are a bit weak. For B-Normal, it only uses per point normal which certainly loses too much positional information on the point cloud. What if it is replaced by a concatenation of both point cloud per-point position and normal? Beyond these two baselines, I’m wondering how it will perform by with 3 branches of input: the acting object, the seed location and scene point cloud.

In table1, why is the result on pushing task significantly lower than other tasks? More explanations on this would be helpful.

Line 288, “These datasets provide 3D mesh reconstruction from 3D scanned point clouds that are captured by 3D sensors. So, here we directly operate on the obtained meshes.” The method requires the holistic scene point cloud as part of the input. In this case, the simulation can be performed during test stage given the acting object and scene’s mesh. The motivation of learning to predict is thus not very clear. When using the partially visible scan as input, the performance is observed to corrupt, as seen in Fig.G.3 fitting on the right, the network predicts all placeable areas as fitting area.


**Reviewer Expertise:**

Very good: Comprehensive knowledge of the area

**Strengths And Weaknesses:**

Strengths:

*The object-object affordance reasoning by self-interaction is novel.

*The code has been provided for reproduction. The code is organized. Its comments and readme file are informative.

Weaknesses:

*In the experiments, the baseline methods of B-Normal and B-Bbox are a bit weak.

*The method requires the holistic scene point cloud as part of the input. In this case, the simulation can be performed during test stage given the acting object and scene’s mesh instead of training a network to do the prediction. The motivation of learning is not very clear.


**Summary Of Recommendation:**

Weak accept. The motivation needs more clarification. Some addtional experiments and analysis are expected.

---

### Meta-Review · Area_Chair_kn7N · 2021-08-13

**Recommendation:** Accept (Poster)
**Confidence:** 5

**Metareview:**

In this paper, an affordance learning framework is proposed for the task of object-object interaction. And a large scale of data are generated in the simulation environment without human labeling. It is recognized that object-object interaction is an important problem for robotic manipulation. While there are some concerns about the performance of the proposed framework when it is applied in real-world scenarios and stronger baselines are preferred.

In the rebuttal session, the authors have responded to the reviewers' concerns and two baselines and more details are provided. And the reviewers have reached a consensus of accepting this paper.

---

### Decision · Program_Chairs · 2021-09-13

**Decision:**

Accept (Poster)

**Comment:**

In this paper, an affordance learning framework is proposed for the task of object-object interaction. And a large scale of data are generated in the simulation environment without human labeling. It is recognized that object-object interaction is an important problem for robotic manipulation. While there are some concerns about the performance of the proposed framework when it is applied in real-world scenarios and stronger baselines are preferred.

In the rebuttal session, the authors have responded to the reviewers' concerns and two baselines and more details are provided. And the reviewers have reached a consensus of accepting this paper.